# Quantifying contact status and the air-breakdown model of charge-excitation triboelectric nanogenerators to maximize charge density

Yike Liu[1,3], Wenlin Liu 📷 [1,3✉], Zhao Wang[1], Wencong He[1], Qian Tang[1], Yi Xi[1], Xue Wang[1], Hengyu Guo[1,2✉] & Chenguo Hu 📷 [1✉]

Surface charge density is the key factor for developing high performance triboelectric nanogenerators (TENG). The previously invented charge excitation TENG provides a most efficient way to achieve maximum charge output of a TENG device. Herein, criteria to quantitatively evaluate the contact efficiency and air breakdown model on charge excitation TENG are established to enhance and evaluate charge density. The theoretical results are further verified by systematic experiments. A high average charge density up to 2.38 mC m$^{-2}$ is achieved using the 4 μm PEI film and homemade carbon/silicone gel electrode in ambient atmosphere with 5% relative humidity. This work also reveals the actual charge density (over 4.0 mC m$^{-2}$) in a TENG electrode based on quantified surface micro-contact efficiency and provides a prospective technical approach to improve the charge density, which could push the output performance of TENG to a new horizon.

[1] Department of Applied Physics, State Key Laboratory of Power Transmission Equipment & System Security and New Technology, Chongqing University, 400044 Chongqing, P. R. China. [2] School of Materials Science and Engineering, Georgia Institute of Technology, Atlanta, GA 30332, USA. [3] These authors contribute equally: Yike Liu, Wenlin Liu. ✉email: liuwl@cqu.edu.cn; cquphysicsghy@126.com; hucg@cqu.edu.cn

The invention of the triboelectric nanogenerator (TENG) provides a most efficient technology of using distributed mechanical energy sources for powering distributed sensor networks in the upcoming internet of things era[1–9]. For TENG devices, the surface charge density, which dominates output performance, is inevitably limited by the air-breakdown effect between two tribo-surfaces[10–15]. In contact-separation mode TENG, derived from Paschen's law and TENG parallel-plate model, the effect of dielectric parameters and atmospheric environment on maximum surface charge density under short-circuit condition has been well expounded[16,17]. These models show that a thinner dielectric film, a larger dielectric constant and a lower atmospheric pressure always lead to a higher maximum surface charge density. For instance, using air ion injection method, the maximum output charge density reaches ~250 μC m$^{-2}$ for 50 μm dielectric, larger than ~150 μC m$^{-2}$ for 125 μm dielectric film in ambient atmosphere[16]. By external charge pumping[18], the maximum surface charge density is enhanced to 490 μC m$^{-2}$ and further to ~1020 μC m$^{-2}$ through reducing the dielectric thickness to 5 μm[19]. Similarly, when pumping the surrounding air pressure to high vacuum (P ~10$^{-6}$ torr) and improving the contact[20], the output charge density for the same TENG device is improved from ~150 μC m$^{-2}$ to ~600 μC m$^{-2}$.

Although surface charge density can indeed be improved by the above strategies, for a common TENG device, the surface charge generated by contact electrification is usually far below the maximum one due to the lack of surface electro-states (electron cloud overlap on the contact interfaces)[16]. In this case, surface modification using physical/chemical methods[21–24], is always adopted for achieving a larger charge output to approach the maximum value. Accordingly, by modifying a 2 μm tribolayer with a biofunctional group, the surface charge density is 200% larger than the one without surface treatment[25]. However, these results are still lower than the maximum value in theory[26–29]. More recently, our group proposed a charge-excitation TENG (CE-TENG)[30], and utilizing the charge produced by external TENG or TENG itself, the main TENG with any dielectric can achieve a maximum charge density of 1.25 mC m$^{-2}$ (the current reported record), which is approximately the maximum value according to the limitation of the air-breakdown effect (Paschen's law). The CE-TENG provides a more effective avenue towards high output. However, with further reduction of the thickness of dielectrics, the insufficient contact caused by surface microstructure highly affects the output. The higher average charge density is hardly obtained due to the insufficient contact. Therefore, it is worth discussing how large the actual charge density in TENG is with a thin dielectric film in insufficient contact level and how to achieve sufficient contact. Unfortunately, there is no method to qualitatively evaluate surface contact level. In addition, an effective air-breakdown model for CE-TENG to reveal the physical mechanism to guide future device development is highly desired.

In this work, we theoretically analyze the maximum charge density of CE-TENG using Paschen's Law and basic capacitor model, and clearly illuminate the strategies of enhancing the charge output for this kind of devices, including the reduction of the thickness of dielectrics, increase of external capacitor and control of atmospheric environment. Corresponding experiments are carried out for further verification. We also find that the contact status between two tribo-surfaces becomes another key point that strongly affects the output performance when dielectric thickness reduces to a few microns. A criterion to qualitatively evaluate contact efficiency is built. To achieve a more sufficient contact, we propose an arched soft contact mode using a homemade carbon/silicone gel electrode, which can improve the contact efficiency from 6.16% to 54.98% for a 4 μm dielectric film.

We achieve the highest record of average charge and energy density up to 2.38 mC m$^{-2}$ and 286.7 mJ m$^{-2}$ by using the carbon/silicone gel electrode and the 4 μm PEI film in ambient atmosphere with 5% relative humidity. This work establishes quantification criteria of surface contact efficiency and a new air-breakdown model for CE-TENG, and provides an effective strategy for evaluating the actual charge density in electrode of TENG and achieving ultra-high output TENG.

## Results

**Maximum charge density of the CE-TENG.** Surface charge density is the key factor for improving the output performance of a TENG device. For a traditional contact-separation mode TENG (Supplementary Fig. 1), with a certain dielectric parameter and atmospheric environment, the surface charge density has a maximum value, which is limited by air-breakdown effect[31,32]. The recent developed CE-TENG provides a most efficient way to achieve the maximum charge density by applying an excitation voltage instead of physical or chemical modification on tribosurface, which significantly improves the working efficiency of TENG. Figure 1a shows the structural schematic of the charge-excitation triboelectric nanogenerator (CE-TENG), which consists of an external TENG (smaller one), a main TENG and a charge-excitation circuit (CEC). Inset 3 presents the photograph of CE-TENG device. Different from the working mechanism of a conventional TENG (Supplementary Fig. 1), the output of CE-TENG is realized by charge transfer between the main TENG and external capacitor[30]. Here, inspired by Xu et al.'s work[33], a half–wave rectifier circuit is used to generate a high excitation voltage, and the corresponding schematic diagram and detailed working process of CE-TENG is shown in Fig. 1b. Firstly, during the charge accumulation process, the charge from the external TENG is excited to the external capacitor through CEC and reaches a certain level after several operation cycles. Then, the charge stored in external capacitor transfers to the main TENG during contact process to reach an equilibrium state, and the charge stored in the main TENG transfers back to the external capacitor correspondingly during the separation process due to the capacitance change of the main TENG when it is pressed and released. Meanwhile, the charge from the external TENG continuously supplements to the external capacitor and thus the main TENG achieves a stable working state. Similar to a conventional TENG, with the rise of charge accumulation, air breakdown would occur between the top electrode and dielectric surface and thus limits the maximum charge output. Figure 1c shows the physical and electric model of the primary part of CE-TENG (including the external capacitor and main TENG). As depicted in Fig. 1d, the potential difference ($V_{gap}$) (simulated results by Comsol Multiphysics) forms between the top electrode and dielectric under a certain air gap. Based on parallel-plate model, the voltage can be derived and expressed by:

$$V_{gap} = \frac{x\sigma}{\varepsilon_0}\left(1 - \frac{x}{\frac{d}{\varepsilon_r} + x + \frac{\varepsilon_0 S}{C}}\right) \quad (1)$$

Where $\sigma$ is the average charge density on the electrode of main TENG when getting contact. $d$ and $\varepsilon_r$ is the thickness and relative permittivity of the dielectric respectively. $\varepsilon_0$, $x$, and $c$ represent the vacuum permittivity, separation gap distance and the capacitance of external capacitor.

On the other hand, according to Paschen's law[34–36], the voltage that causes air breakdown between two parallel plates complies with

$$V_{a-b} = \frac{A(Px)}{\ln(Px) + B} \quad (2)$$

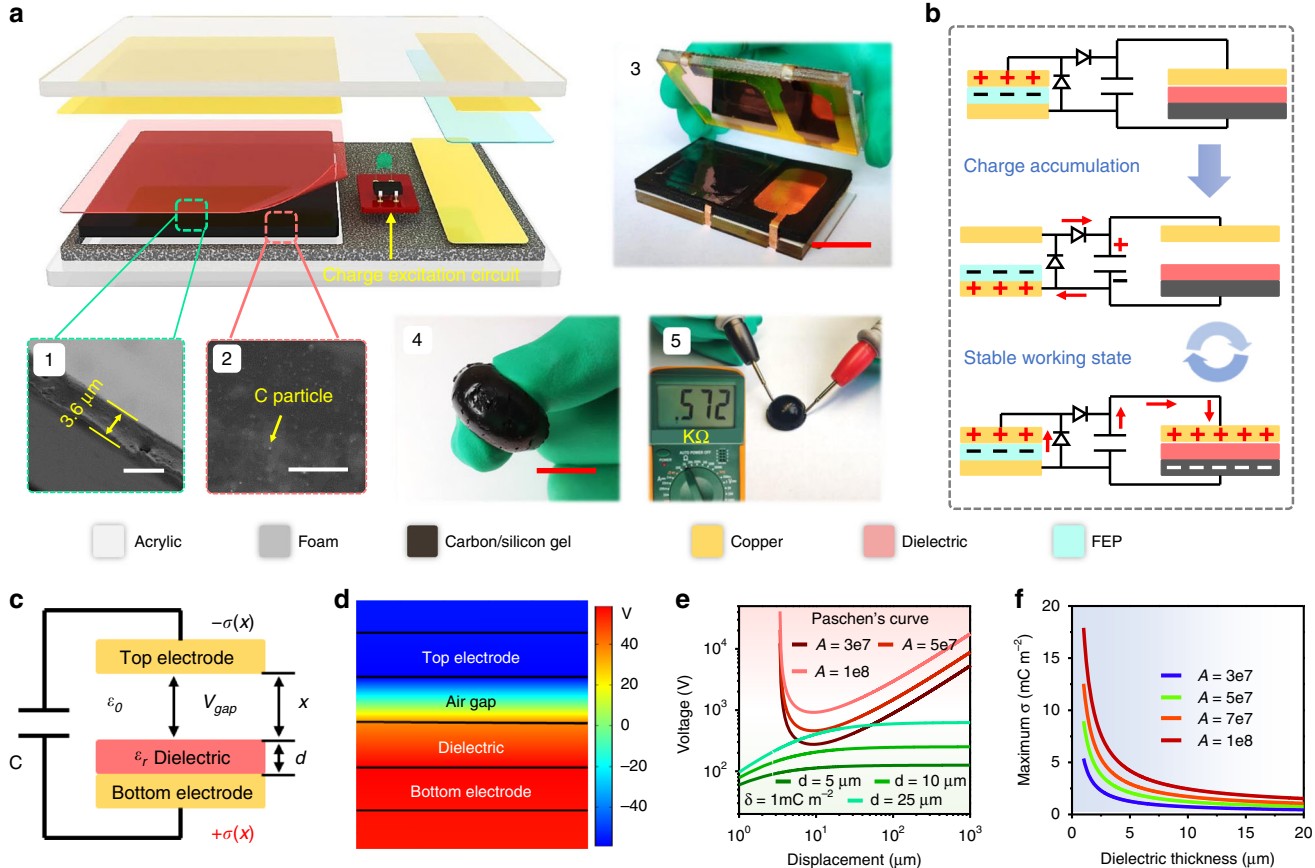

**Fig. 1 Working mechanism and maximum charge density. a** Structural schematic of charge-excitation triboelectric nanogenerator (CE-TENG). Inset 1 shows cross-section view (SEM) of the dielectric film (scale bar: 5 μm). Inset 2 is the SEM image of carbon nanoparticle dispersed in silicone matrix (scale bar: 20 μm). Inset 3 presents the photograph of CE-TENG device (scale bar: 2 cm). Inset 4 is the photograph of carbon gel electrode. Inset 5 demonstrates the conductivity of carbon gel electrode (scale bar: 1 cm). **b** Basic working mechanism of CE-TENG. **c** Equivalent physical and electric model of CE-TENG and some critical parameters are also listed in the schematic. **d** Simulated potential distribution of main TENG under a certain air gap distance using Comsol Multiphysics (electrodes and dielectric thickness: 0.1 mm, air gap: 0.1 mm, dielectric constant: 4.6, $\sigma(0) = 2 \times 10^{-3}$ C/m$^2$). **e** Giving the theoretical Paschen's curves under different value of A which determined by atmospheric content, and voltage curves between top electrode and dielectric with various dielectric thickness under a certain charge density $\sigma(0) = 1$ mC m$^{-2}$. **f** The maximum charge density limited by air breakdown with different dielectric thickness and atmospheric constant A.

Where $P$ is the barometric pressure. $x$ is the gap distance between two plates. $A$ and $B$ are the constants determined by atmosphere, including atmospheric composition, relative humidity, temperature, and etc. $A$ is inversely proportional to the relative humidity and proportional to maximum charge density. For a common ambient condition, $A$ equals $2.87 \times 10^7$ V (atm·m)$^{-1}$, and $B$ equals 12.6.

To avoid air breakdown, it requires that $V_{a-b}$ is always larger than $V_{gap}$. According to Eqs.(1) and (2), Paschen's curves under different values of $A$ and voltage curves between the top electrode and dielectric with various dielectric thickness under a certain charge density $\sigma(0) = 1$ mC m$^{-2}$ are plotted as shown in Fig. 1e. The plots reveal that the adjustment of atmospheric constant $A$ (other than regulating of air pressure) and the dielectric thickness can prevent two curves from intersecting. In this case, the maximum charge density for different dielectrics and ambient conditions can be deduced as:

$$\sigma_{\max} = \left( \frac{AP\varepsilon_0}{(\ln(Px) + B)\left(1 - \frac{x}{\frac{d}{\varepsilon_r} + x + \frac{\varepsilon_0 S}{C}}\right)} \right)_{\min} \quad (3)$$

As depicted in Fig. 1f, with the decrease of dielectric thickness and

increase of atmospheric constant A, the maximum charge density can be largely improved, which provides an enlightening strategy towards ultra-high output CE-TENG. The detailed deriving steps and discussions are presented in Supplementary Note 1.

It is worth noting that, with further reducing the thickness of dielectric, the contact efficiency would become worse and thus diminish the real charge output (Supplementary Note 2 and Supplementary Table 1). In this work, we adopt the 4 μm PEI film (inset 1, all the dielectric films used in the experiment are commercially obtained and 4 μm PEI film is the thinnest one we can purchase from the market) and soft carbon/silicone gel electrode to improve the contact efficiency (inset 4, the conductivity of the carbon/silicone gel electrode is about 2.5–7.6 × 10$^{-3}$ Ω·m). Inset 2 and 3 are the SEM images of the carbon/silicone gel electrode material and inset 5 shows its conductivity. The carbon/silicone gel electrode is shapeable and soft (Supplementary Movie 1), which can realize conformal contact with rough surface. The fabrication of the carbon/silicone gel electrode is displayed in Methods and Supplementary Fig. 2.

**Critical factors that effect on the real output charge density.** For CE-TENG, the dielectric thickness and contact status would affect the contact capacitance of the main TENG device, which

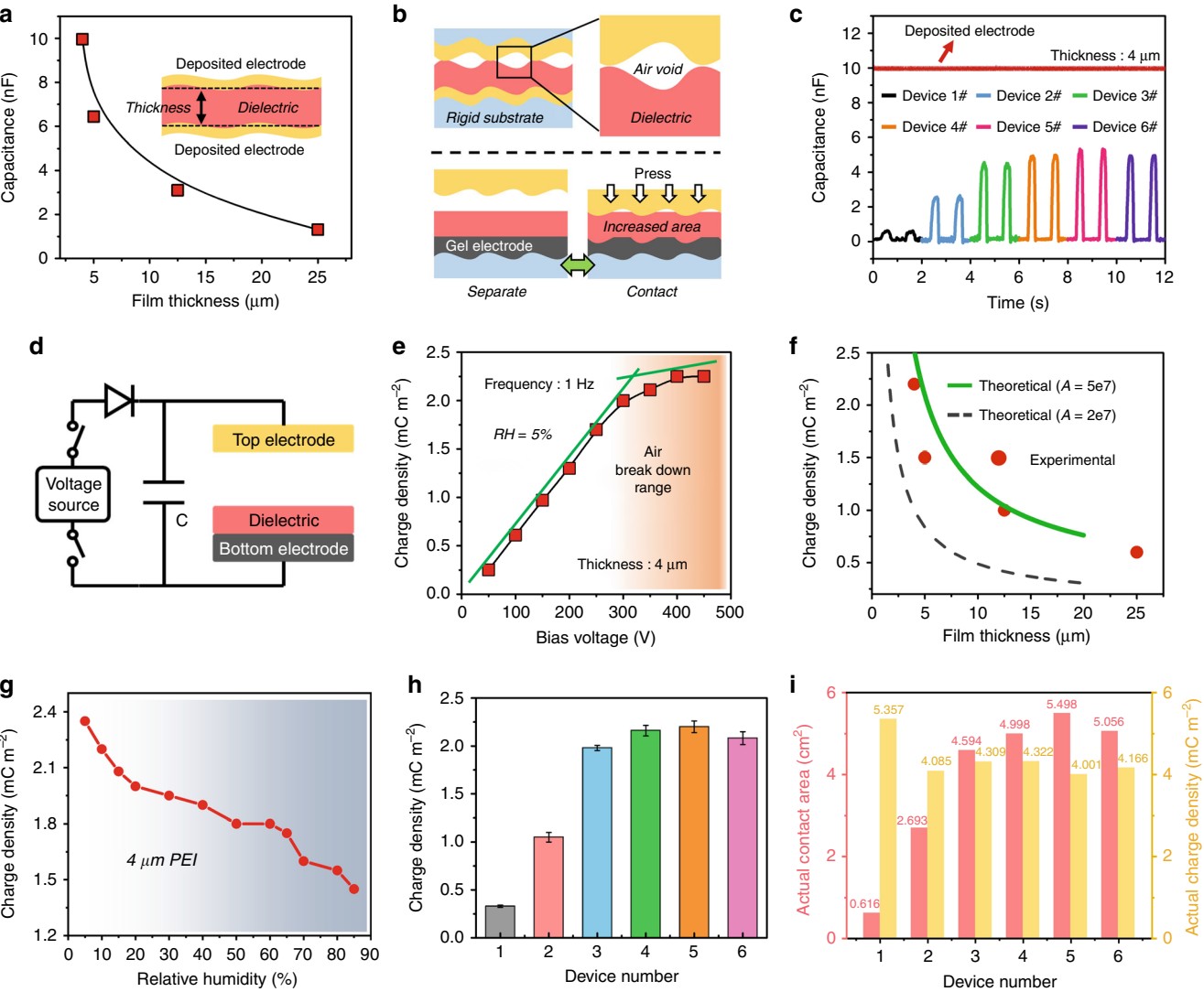

**Fig. 2 Critical factors that effect on the real output charge density. a** Capacitance of the capacitor made of dielectrics with different thickness (by depositing electrode, electrode area: 3.2 × 3.2 cm). **b** The schematic of contact status when using a rigid electrode (up) and soft gel electrode (down). **c** The capacitance of main TENG devices with six contact optimization compared to the one with deposited electrode (regarded as fully sufficient contact; dielectric thickness: 4 μm). **d** Electric scheme of CE-TENG when exciting by a voltage source. **e** Output charge density when applying different bias voltage (operation frequency: 1 Hz, relative humidity: 5%, dielectric thickness: 4 μm). **f** Experimental and theoretical charge density of CE-TENG with different dielectric thickness. **g** Output charge density of CE-TENG when varying atmospheric humidity (dielectric thickness: 4 μm PEI, bias voltage: 400 V). **h** Output charge density of CE-TENG with different contact status (dielectric thickness: 4 μm, bias voltage: 350 V). **i** The actual contact area and actual charge density of main TENG devices with six contact optimization (electrode area: 10 cm², dielectric thickness: 4 μm).

determines the maximum charge quantity that can be stored in TENG and transferred to the external capacitor. By depositing Cu (electrode) on both sides of the dielectrics using vacuum evaporation (this status can be regarded as 100% contact efficiency of TENG device), the capacitance of dielectrics with different thickness are measured and shown in Fig. 2a (the detailed photograph and data are shown in Supplementary Fig. 3), where the results are highly consistent with the parallel-plate capacitance determined by Eq. 4.

$$C = \frac{\varepsilon_r S}{4\pi k d} \qquad (4)$$

In practical TENG devices, there would be large air voids existence due to the surface microstructure of the electrode, dielectric and substrate materials when the top and bottom part contacts (shown in upper of Fig. 2b). In Supplementary Fig. 4, by depositing Cu on the sandpaper with various roughness as

electrodes, the charge density increases with the reduction of roughness, proving the higher charge density achieved on smoother contact surface, which is different to the common TENG that rougher surface contributes to higher charge density. In this work, we propose and fabricate a carbon/silicone gel electrode to form an ultra-soft contact status, which can be easily conformal contact with the dielectric film and highly increase the contact efficiency (shown in the lower of Fig. 2b). To improve the contact efficiency, we further introduced the arched electrode structure with different parameters (shown in Supplementary Fig. 5). Figure 2c shows the capacitance of main TENG devices with six contact optimizations (Supplementary Table 2) versus the one with deposited electrodes. Here, we define the contact efficiency as:

$$\eta = \frac{C_{contact}}{C_0} \qquad (5)$$

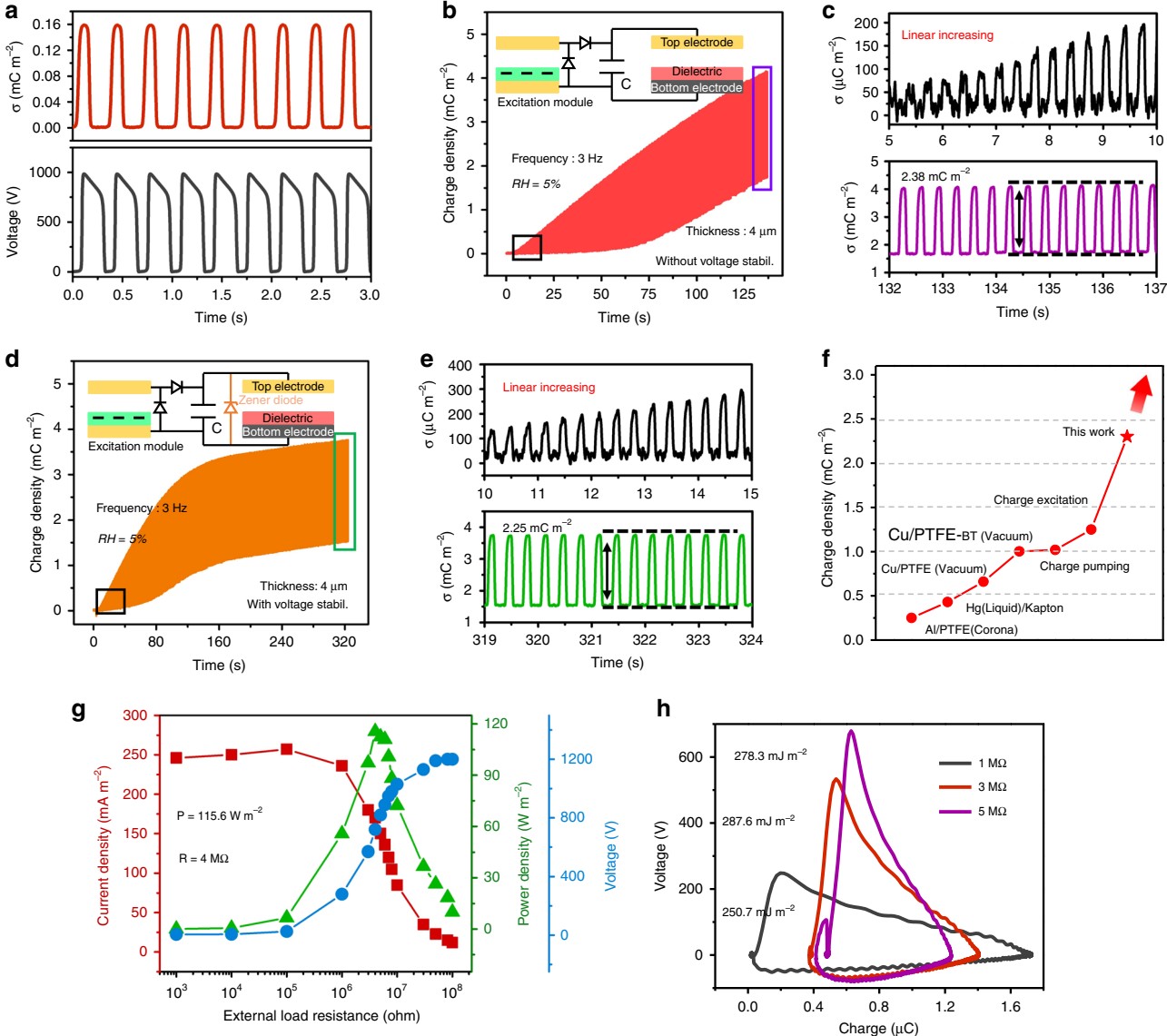

**Fig. 3 Electric output of CE-TENG with optimized materials and structure. a** Output charge and voltage curve of external excitation TENG (operation frequency: 1 Hz). **b** Dynamic charge density output of CE-TENG without using zener diode. **c** Enlarged initial charge accumulation curve (up) and saturation output state (down). **d** Dynamic charge output of CE-TENG with using zener diode. **e** Enlarged initial charge accumulation (up) and saturation output state (down). **f** Output charge density comparison in the development stage of TENG. **g** Output current, voltage and peak power under various external load (operation frequency: 3 Hz). **h** V–Q curve of CE-TENG under optimized load resistance (operation frequency: 3 Hz).

Where $C_{contact}$ is the capacitance of the main TENG in contact state, and $C_0$ is the capacitance of the dielectric with deposited electrodes (Fig. 2a). In the experiments, through our contact optimization, the contact efficiency of 4 μm dielectric can be improved from 6.16% to 54.98% as listed in Supplementary Table 3.

For the sake of simplicity, we first use external DC voltage source to provide an excited voltage in order to systematically investigate the effect of dielectric thickness, contact optimization, and atmospheric environment on the output charge of CE-TENG, as shown in Fig. 2d. In theory, a larger external capacitor would lead to a higher charge transfer (Eq. 3). Supplementary Fig. 6a demonstrates the impact of different external capacitance on the output charge, indicating that larger external capacitor leads to higher charge density. However, a large external capacitor spends a longer time to reach charge saturation as shown in Fig. 3b and Supplementary Fig. 6b. Therefore, on balance, we adopt 25nF external capacitance to test the entire output

performance characteristics. The output charge density of CE-TENG with 4 μm dielectric under a constant environment and application of different excitation voltages are shown in Fig. 2e. With the voltage raising and reaching over 400 V, the output charge density linearly increases and gets the saturated due to the limitation of air-breakdown effect discussed above. In this way, the output charge with different dielectric thickness is measured under relative humidity of 5% as shown in Fig. 2f (red dot). The SEM of dielectric surface and cross-section view are demonstrated in Supplementary Fig. 7. Compared with previous works (dash line)[18,19,30,37,38], the maximum charge density holds a certain improvement. The possible reason is that the previous theoretical and experimental results were both obtained in the relative humidity around 40–70%, which would have high impact on the atmospheric constant $A$ and thus affect the air breakdown of TENG. By adjusting the constant $A$ from 2E7 to 5E7, the experimental results get well matched with the theory (green solid line). In addition, the measured results of atmospheric humidity

on the output charge density are shown in Fig. 2g, from which, with the increase of relative humidity, the output charge density decreases as expected by air-breakdown model. Finally, with an optimized contact status, average output charge density of ~2.2 mC m$^{-2}$ can be achieved using the 4 μm PEI dielectric under relative humidity of 5% (Fig. 2h).

Here, considering the incomplete contact in TENG, which means that the actually effective contact area is smaller than the area of electrode (S = 10 cm$^2$), inspired by the contact efficiency above, we can get the actual effective contact area $S'$ and actual charge density $\sigma'$ in TENG.

$$S' = S \cdot \frac{C_{contact}}{C_0} = S \cdot \eta \tag{6}$$

$$\sigma' = \frac{Q}{S'} = \frac{\sigma}{\eta} \tag{7}$$

Where Q is the measured charge quantity and $\sigma$ is the average charge density. The detail discussion is given in Supplementary Note 3.

From the above discussion, the criterion to qualitatively evaluate the contact efficiency is built. The actual contact area and the actual charge density of the CE-TENG with six contact levels are shown in Fig. 2i (electrode area: 10 cm$^2$). Obviously, the contact level has large influence on average charge density and the output charge quantity depends on the actual charge density and actual contact area, but the actual charge density of CE-TENG is basically the same under different contact levels. The actual charge density in CE-TENG is over 4.0 mC m$^{-2}$, which also means that the maximum output charge density could be up to 4.0 mC m$^{-2}$ in an ideal surface contact level. Therefore, reducing dielectric thickness, increasing that the external capacitor, improving contact efficiency and controlling ambient condition all benefit the charge output enhancement.

**Output performance of the CE-TENG**. With the optimized condition, the output performance of CE-TENG using external TENG as charge source should be systematically investigated. Figure 3a exhibits the basic short-circuit charge and open-circuit voltage output of the external TENG. According to Fig. 2e, the voltage over 1000 V is enough to ensure the main TENG working near the maximum level. In Fig. 3b–e, the dynamic charge accumulation of CE-TENG under 3 Hz frequency without (Supplementary Movie 2) and with (Supplementary Movie 3) using zener diode[25,30] are recorded, respectively. The voltage across the external capacitor is also measured and presented in Supplementary Fig. 8, which reveals the stable voltage around 450 V. From the enlarged initial and final process (Fig. 3c, e), the charge accumulates linearly and finally reaches saturation to ~2.38 mC m$^{-2}$ (without zener diode) and ~2.25 mC m$^{-2}$ (with zener diode) (the calculated effective charge density are shown in Supplementary Fig. 9). This experimental result achieves the highest record of charge output density in the TENG history. And following the instructive strategies above, the higher output could be further expected as summarized in Fig. 3f. In addition, the peak power and energy output per cycle of CE-TENG is measured under 3 Hz low working frequency with varied external load resistance. From Fig. 3g, h, with the matched impendence 4 MΩ and 3 MΩ, the ultra-high peak output and energy can reach 115.6 W m$^{-2}$ and 286.7 mJ m$^{-2}$, respectively.

**Demonstrations of the CE-TENG**. To demonstrate the output performance of CE-TENG in practical applications, we first measure the short-circuit current and voltage (the voltage is measured from a 10 MΩ load, which is connected with external

capacitor and main TENG in series to prevent the electric breakdown of dielectric film) under 5 Hz working frequency as depicted in Fig. 4a, b, respectively, (output under other frequencies are shown in Supplementary Fig. 10). The peak current and voltage can reach up to ~360 mA m$^{-2}$ and ~1200 V, which can be directly used for powering 456 LEDs with diameter of 5 mm to high brightness in series (Fig. 4c and Supplementary Movie 4). Afterwards, with the use of full-wave rectifier (Fig. 4d), the CE-TENG can charge 2.2 μF capacitor to 200 V (Supplementary Movie 5) at 3 Hz. Under the operation frequency of 3 Hz, the equivalent charging rate reaches as high as 13.6 μC s$^{-1}$ (Fig. 4e) when charging 1 mF and 470 μF capacitors. And it can be used for driving a commercial thermo-hygrometer in sustainable working mode (Fig. 4f and Supplementary Movie 6) at 5 Hz. Finally, we test the stability during 30k operation cycles as shown in Fig. 4g. The charge density is gradually decreasing at the early stages and then tends to be stable. Supplementary Note 4 and Supplementary Fig. 11 discuss the reason of possible output decreasing mechanism of CE-TENG.

## Discussion
In summary, a criterion to qualitatively evaluate the surface contact level between an electrode and dielectric (contact efficiency) is established for TENG. A new air-breakdown model on charge-excitation TENG is built based on Paschen's law, and we achieve a high charge density by multiple parameter adjustment, including reduction of the thickness of dielectrics, increase of the external capacitor, control of the atmospheric environment and optimization of surface contact level. We also demonstrate that the previous defined charge density belongs to average charge density due to the incomplete contact, an actual charge density in TENG is obtained based on micro-contact efficiency theory. The actual charge density in CE-TENG is over 4.0 mC m$^{-2}$, which also means that the maximum output charge density could be up to 4.0 mC m$^{-2}$ in an ideal surface contact level, which would have a great significance in evaluating the actual charge density on the surface of dielectric materials for TENG. Besides, a soft carbon/silicone gel electrode is invented to improve contact efficiency from 6.16% to 54.98% for a 4 μm dielectric film, by which the high average charge and energy density of 2.38 mC m$^{-2}$ and 286.7 mJ m$^{-2}$ are obtained on CE-TENG in ambient atmosphere with 5% relative humidity. This work provides a new insight into the way of enhancing the output performance, which could push triboelectric nanogenerators to a new stage.

## Methods
**Fabrication of the soft carbon silica gel electrode**. First, the 4 mm thick foam was cut with the dimensions of 67.2 × 44.4 × 4 mm$^3$ (Marked as foam 1), and the 0.8 mm thick foam was cut with the dimensions of 67.2 × 44.4 × 0.8 mm$^3$ (Marked as foam 2). Then, making a square groove with the dimensions of 32 × 32 × 0.8 mm$^3$ on the right side of the foam 2. The foam 2 with a square groove was adhered to the foam 1. Required amounts of Parts A and B of Ecoflex 20 were dispensed into a Petri dish (1 A: 1B by volume and were stirred fully for 3 min.) An appropriate amount of fine carbon powder (Tanfeng Tech) was put into the mixture, which was stirred for 3 min. Then, the coarse carbon powder was mixed into the mixture with fine carbon powder and silica gel. After stirring for 4 min, the mixture was poured into the square groove, which was sandwiched between two acrylic based plates and pressed for 8 h. At this point, the fabrication of the silica gel electrode was finished. The fabrication process is shown in Supplementary Fig. 2a.

**Fabrication and integration of the TENG**. As is shown in Fig. 1a, the CE-TENG is mainly composed of two parts. For one part, it consists of an acrylic based plate and two Cu electrodes. The fabrication process of this part is as follows. First, two Cu electrodes were attached to the left and right sides of the acrylic plate, which was cut by laser cutter with dimensions of 67.2 × 44.4 × 4 mm$^3$. And the size of two Cu electrodes was 32 × 32 mm$^2$ and 32 × 15.0 mm$^2$, respectively. 44 mm ×20 mm × 12.5 μm FEP film was attached to the upper surface of 32 × 15.0 mm$^2$ Cu electrode at last. For another part, 67.2 × 44.4 × 1 mm$^3$ arch structure that was made by 3D

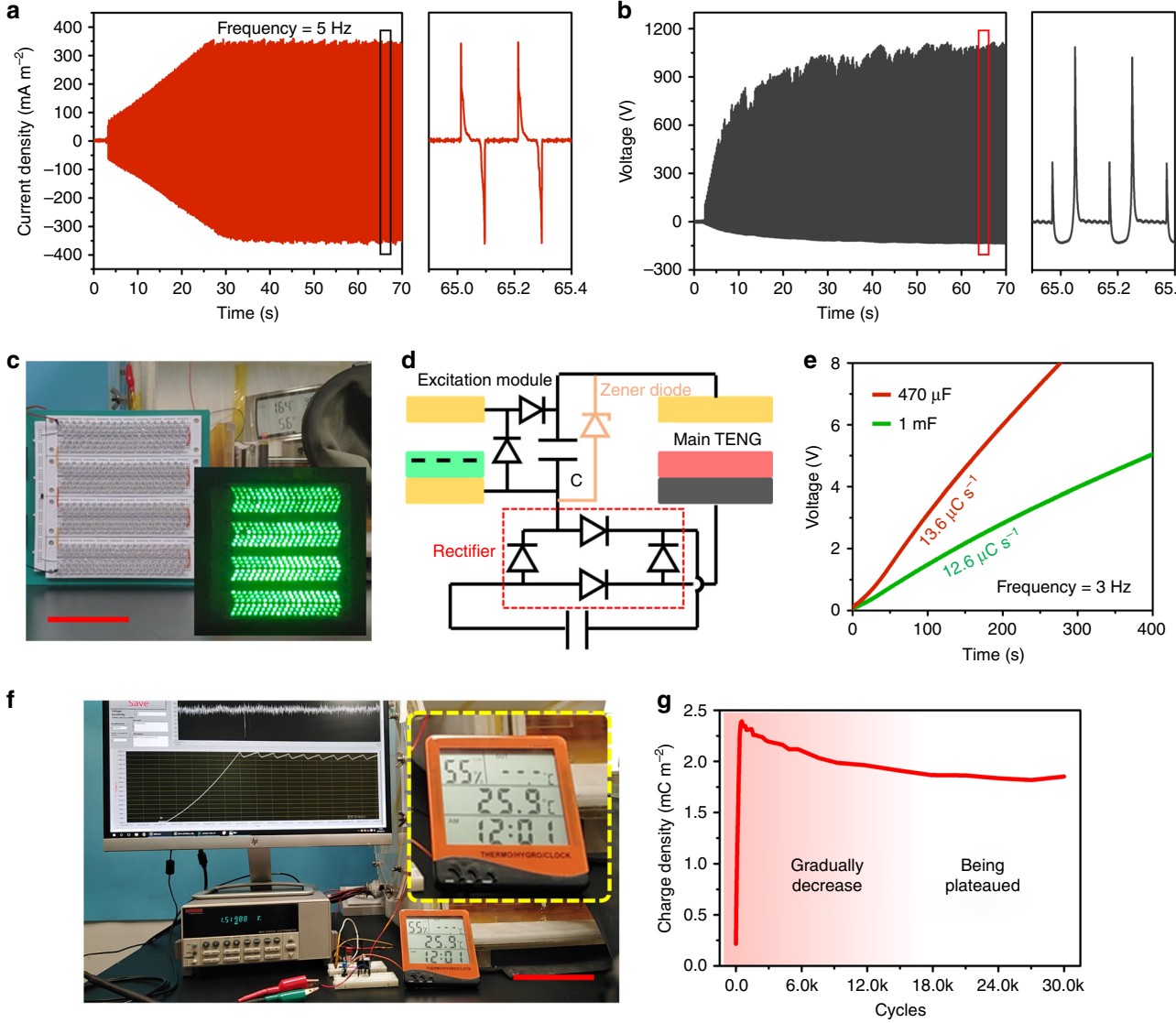

**Fig. 4 Demonstrations of high output CE-TENG. a** Current density and **b** voltage output of CE-TENG under 5 Hz operation frequency. **c** Directly powering hundreds of LEDs by CE-TENG (scale bar: 10 cm). **d** Electric scheme of CE-TENG for charging capacitor. **e** Charging curve of 1 mF and 470 μF capacitor using CE-TENG (frequency: 3 Hz). **f** CE-TENG for powering thermometer under 5 Hz operation frequency (scale bar: 10 cm). **g** Long-term stability test of CE-TENG.

printer was attached to the upper surface of $67.2 \times 44.4 \times 4$ mm$^3$ acrylic based plate. Then, $32 \times 15.0$ mm$^2$ Cu electrode was attached to the left side of the foam with silica gel electrode, which was adhered to the upper surface of the arch structure. Finally, 44 mm × 44 mm × 4 μm PEI film was attached to the surface of the silica gel electrode.

**Fabrication of the evaporation coating electrodes**. A dielectric film was sandwiched between two acrylic frames, the shape of the frame is the same as the shape of the electrode. The prepared devices were put into the high vacuum evaporation coating machine. Utilizing vacuum evaporation technology Cu was deposited on the both sides of the dielectric film as the electrodes. The fabrication process of the sandpaper electrodes mainly consists of three parts. Firstly, $44 \times 44$ mm$^2$ sandpaper was attached to the left side of the $67.2 \times 44.4 \times 4$ mm$^3$ acrylic based plate, $67.2 \times 44.4 \times 1$ mm$^3$ acrylic frame with a $32 \times 32 \times 1$ mm$^3$ square groove was adhered to the surface of the sandpaper. Then, the device was put into the high vacuum evaporation coating machine. Finally, taking out the device and removing the acrylic frame, the sandpaper electrode was finished.

**Electric measurement and characterization**. The measurement of the electrical performance of TENG was carried out in a $50 \times 50 \times 95$ cm$^3$ acrylic glove box. The contact-separation process of TENG was driven by a linear motor (WEI-NERMOTOR WMU-090-D). The humidity of the glove box was controlled by the reusable silica gel desiccant, and the reuse of the silica gel desiccant was realized by

a freeze dryer (Bilon FD-1B-80). The temperature was controlled by a constant temperature circulating water tank (HX-105) and homemade copper tubes with a blower. The voltage of the capacitor was measured by electrostatic voltmeters (Trek 370). The transferred charges and the short-circuit current were measured by an electrometer (Keithley 6514). The capacitance of the device with sinusoidal motion was measured by a 7–1/2 digital graphical sampling multimeter (Keithley DMM7510). A source meter (Keithley 6517) was adopted as the voltage source for providing excitation voltage. The temperature and humidity were measured by a digital temperature humidity atmospheric pressure gauge (Testo 622), the relative humidity was controlled in the range of 4–6%, and the temperature was controlled in the range of 16–17 °C, The Cu electrodes on both sides of the dielectric film are plated by high vacuum evaporation coating machine (VZZ-300). The scanning electron microscope images were taken by scanning electron microscopy (SEM, TESCAN VEGA 3 SBH SEM).

## Data availability
The data that support the plots within this paper and other findings of this study are available from the corresponding authors upon reasonable request.

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

## Acknowledgements

This work was supported by National Natural Science Foundation of China (NSFC) (51572040 and 51772036), and the Fundamental Research Funds for the Central Universities (2019CDXZWL001, 2018CDQYWL0046, and 2018CDPTCG0001/22).

## Author contributions

C.H. supervised the project. W.L., Y.L. C.H. and H.G. conceived the project and designed the experiment part. Y.L. fabricated the devices and completed the electrical performance measurement. Z.W., W.H., Q.T., Y.X. and X.W. helped plot and analyze data. Y.L., H.G., W.L. and C.H. wrote the manuscript. All authors contributed to the manuscript.

## Competing interests

The authors declare no competing interests.
