## [Peer Review File · Nature Communications]

Reviewers' Comments:

Reviewer #1:

Remarks to the Author:

This manuscript reports an advanced approach to enhance and evaluate the surface charge density of triboelectric nanogenerator (TENG), which is vitally important in TENG related energy harvesting and sensing. The authors demonstrated the highest record of average charge density up to 2.38 mC m⁻² using a 4 μm PEI film and homemade carbon/silicone gel electrodes with an ambient atmosphere of 5% relative humidity. The prospective technic approach provided in this manuscript for surface charge density enhancement, will greatly push the output performance of TENG to a new horizon. The outcome of this work is impressive and is also important in the TENG field. Thus, I recommend that this manuscript can be accepted with a minor revision.

1: Several works have been reported for boosting the surface charge in TENG. What is the major difference between this work and the existing work about the surface charges? You can explain them in the introduction component.

2: The green color in Figure 1b is not explained in the manuscript. Please note the meaning of the colors in the figures or somewhere. If not, it might confuse the readers.

3: Please note the specific information about COMSOL simulation in Supporting information (Figure 1d). For example, the parameters and the model of the simulation.

4: Why the relative humidity (Figure 2g) is important to the output of TENG? Please explain in the manuscript.

5: Please provide more details about how to measure the open circuit voltage, short circuit current and transferred charge (e.g. connection of electrodes).

6: What is the difference between Figures 3c and 3e? Please explain the purpose of them.

7: The voltage curves in Figure 3a and Figure 4b look different. Were you using different equipment for the measurement?

8: Please double check the conclusion component. Several "space" need to be deleted in this component. For example, line 263 and line 267.

Reviewer #2:

Remarks to the Author:

This review relates to the Manuscript Number NCOMMS-19-195864 submitted to "Nature Communications" journal entitled, "Quantification of surface contact efficiency and air breakdown model on charge excitation TENG for maximum charge density."

The manuscript reports on an interesting strategy to achieve high charge density based on Paschen's law, which is intended to improve the output performance of TENGs. While premise is certainly interesting, there are major questions that are not fully addressed, and publications at this stage seems a bit premature.

1. Given that the optimized thickness of PEI, 4 μm, it is not clear how this is achieved and/or controlled. Authors highlighted the contact efficiency, but there is a lack of academic/scientific description or understandable mechanism that how this thickness is associated. It would be interesting to show thickness (<4 μm) dependence with comments about this.

2. Authors particularly stated that "In the experiments, through our contact optimization, the contact efficiency of 4 μm dielectric can be improved...", but it is not at all clear why authors characterized the whole system taking advantage of the external

capacitance 25 nF, and the explanation provided is far from adequate.

3. It is not particularly useful to compare (in Fig. 2e) charge density as a function of bias voltage (DC) and derived charge output values generated by charge excitation TENG (AC or pulsed AC), which is not fully accurate.

4. What accounts for the varied charge density as a function of relative humidity in Fig. 2g? Has there been any charge density drop in optimized PEI? This seems like far more important parameter that has not been discussed at all. One of the weakest parts of the manuscript is in the characterization of the device at 5% relative humidity (RH). I would suggest a few more experimental test to be performed at different RH (moderate level).

5. Authors studied the performance of the CE-TENG at 3 Hz of applied mechanical load, yet demonstration presenting its high output performance was measured at 5 Hz. How working frequency would be expected to function at their capacitance of the whole device?

On a more general note, the manuscript seems fall short of the standards expected from a journal like Nature Communications. Unfortunately, based on the above comment, I am unable to recommend publication.

Reviewer #3:

Remarks to the Author:

The charge density is one of the key indicators to evaluate triboelectric nanogenerator's performance. It has been improved from 460 $\mu\text{C}/\text{m}^2$ (Nat. Communication, 2018, DOI: 10.1038/s41467-018-06045-z) to 1000 $\mu\text{C}/\text{m}^2$ (Nano Energy, 49, 625-633 (2018)) to 1250 $\mu\text{C}/\text{m}^2$ (Nat. Communication, 10, 1426 (2019)). This paper presents an interesting experimental and theoretical investigation on further improving charge density in a charge excitation triboelectric nanogenerator (TENG). By using a 4 μm thick PEI film and a homemade carbon/silicon gel electrode, the authors managed to push the charge density to 2.38 mC/m^2 , setting a new record for TENGs. Theoretical modeling predicts an even higher value. This research is of substantial interest to the scientific communities in energy harvesting devices, material science and device physics. All figures are well prepared and the experimental data are carefully analyzed. I recommend to publish this paper.

A few more comments/questions:

1. How to set the optimum Zener diode in advance, as the offset in Zener value might deviate the optimum power?
2. Can the authors comment on the device performance (charge accumulation, device stability) at frequencies higher than 5 Hz?
3. Has the energy conversion efficiency of the reported CE-TENG been measured? If yes, what is it? Is it following the same trend as that of the charge density?
4. In supporting Figure 6, the authors report a maximum charge density of 2.55 mC/m^2 that was achieved when using an external storage capacitor of 110 nF. This value is better than the one reported in the abstract. Shall this 2.55 mC/m^2 be mentioned in the abstract or in the conclusion?
5. Can the authors provide more information about the electric properties of their homemade carbon/silicon gel electrode?
6. Two typos:
 - a. Line 45: "concatct"
 - b. Line 55: "(CE-ENG)"

Point-to-Point Response to the Reviewer's Comments
(Comments in black, response in blue)

Dear reviewers:

Thank you for your detailed and useful comments and suggestions on our manuscript. We have revised the manuscript accordingly and the detailed corrections are listed below point by point.

Reviewer #1 (Remarks to the Author):

This manuscript reports an advanced approach to enhance and evaluate the surface charge density of triboelectric nanogenerator (TENG), which is vitally important in TENG related energy harvesting and sensing. The authors demonstrated the highest record of average charge density up to 2.38 mC m⁻² using a 4 μm PEI film and homemade carbon/silicone gel electrodes with an ambient atmosphere of 5% relative humidity. The prospective technic approach provided in this manuscript for surface charge density enhancement, will greatly push the output performance of TENG to a new horizon. The outcome of this work is impressive and is also important in the TENG field. Thus, I recommend that this manuscript can be accepted with a minor revision.

Answer: We highly appreciate the reviewer's positive comments on our work as "impressive". And we also thank the reviewer's detailed and responsible reviewing of our work.

1. Several works have been reported for boosting the surface charge in TENG. What is the major difference between this work and the existing work about the surface charges? You can explain them in the introduction component.

Answer: Thank the reviewer for this good suggestion. We have added the related information in the introduction component. Previous works mainly concentrated on surface/inner material modification and controlling working environment (high vacuum $\sim 10^{-4}$ Pa). Although surface charge density can indeed be improved by the above strategies, for a common TENG device, the surface charge generated by contact electrification is usually far below its theoretical maximum value. In this work, we utilize our recently developed charge excitation TENG (CE-TENG) which can ensure the device working on its maximum state, and analyze the surface charge density model using Paschen's Law which indicates that a thinner dielectric layer would lead

to a higher output charge density in CE-TENG. Furthermore, in the experiment, we found that when reducing the thickness of dielectric, the contact efficiency between top electrode and dielectric layer became a significant aspect. Following the theoretical analysis and experimental optimization, we adopted the ultrathin 4 μm PEI film and homemade carbon/silicone gel electrode in ambient atmosphere with 5% relative humidity. The ultrathin dielectric film and flexible contact make the CE-TENG achieve the maximum output charge density. And the detailed comparison has shown in introduction part.

2. The green color in Figure 1b is not explained in the manuscript. Please note the meaning of the colors in the figures or somewhere. If not, it might confuse the readers.

Answer: Thanks for your detailed reviewing and suggestion. We have conformed colors in figure 1b to 1a.

3. Please note the specific information about COMSOL simulation in Supporting information (Figure 1d). For example, the parameters and the model of the simulation.

Answer: Thank you for this valuable suggestion. The corresponding parameters of the simulation have been added in the figure caption.

electrodes and dielectric thickness: 0.1 mm, air gap: 0.1 mm, dielectric constant: 4.6,

$$\sigma(0) = 2 \times 10^{-3} \text{C/m}^2$$

4. Why the relative humidity (Figure 2g) is important to the output of TENG? Please explain in the manuscript.

Answer: Thanks for your good question. We have explained this question in the manuscript. According to the Paschen's law and charge excitation TENG (CE-TENG) mode, the maximum charge density can be deduced as (Supplementary Note 1):

$$\sigma_{max} = \left(\frac{AP\epsilon_0}{(\ln(Px)+B) \left(1 - \frac{x}{\frac{d}{\epsilon_r} + x + \frac{\epsilon_0 S}{C}} \right)} \right)_{min} \quad (1)$$

Where A is the constant determined by atmosphere, including atmospheric composition, relative humidity, temperature and etc. A is inversely proportional to the relative humidity and is proportional to maximum charge density, therefore, we can enhance maximum charge density by setting low relative humidity environment to enhance A as shown in Figure 1f and Figure 2g from theory and experiment aspects.

Fig. 1f | The maximum charge density limited by air break down with different dielectric thickness and atmospheric constant A. Fig. 2g | Output charge density of CE-TENG when varying atmospheric humidity (dielectric thickness: 4um PEI, bias voltage: 400 V)

5. Please provide more details about how to measure the open circuit voltage, short circuit current and transferred charge (e.g. connection of electrodes).

Answer: Thanks for your good suggestion. The transferred charge and short-circuit current were measured by Electrometer (Keithley 6514), and the voltage was measured by Electrostatic Voltmeters (Trek 370). In principle, the electrometer is connected to circuit in series when measuring charge/current, and the electrostatic

voltmeters is connected in parallel with load when measuring voltage. For a common TENG, Figure R1a shows related circuit diagram to measure charge/current/voltage for common TENG with load, and the short-circuit charge/current and open-circuit voltage of common TENG are shown in Figure R1b and c respectively. The measuring circuit of short-circuit charge/current and voltage of charge excitation TENG in this work are shown in Figure R2. Owing to the super-high charge density and very thin dielectric film, on the condition of open circuit, the voltage between the two electrodes in main TENG will break down the dielectric film and damage the device, thus, we measure the voltage on a 10 M Ω load (Figure R2b) to evaluate the output voltage of CE-TENG, and we have corrected the description on voltage measurement in the manuscript.

Figure R1. Electric scheme of common TENG for testing charge, current and voltage.

Figure R2. Electric scheme of CE-TENG for testing short-circuit charge/current **a** and voltage **b**.

6. What is the difference between Figures 3c and 3e? Please explain the purpose of them.

Answer: Thanks for your good question. Figures 3c is the output charge density of CE-TENG without using zener diode and Figures 3e is output charge density with using zener diode. They are the enlarge image of Figures 3b and 3d respectively. From Figures 3c and 3e, we can clearly see that the charge accumulates linearly and finally reaches saturation to $\sim 2.38 \text{ mC m}^{-2}$ (without zener diode) and $\sim 2.25 \text{ mC m}^{-2}$ (with

zener diode). The purpose of these two figures is to compare the output performance of CE-TENG without zener diode and with zener diode. The experiment result shows that the output charge density with voltage stabilization is more stable than the charge density without voltage stabilization.

7. The voltage curves in Figure 3a and Figure 4b look different. Were you using different equipment for the measurement?

Answer: Thanks for your good question. They are both measured by an Electrostatic Voltmeters (Trek 370; Rang: $0 - \pm 3KV$). However, Figure 3a shows the open-circuit voltage curves of excitation TENG with a half rectifier circuit under 3 Hz working frequency and the half rectifier circuit is used to generator a high voltage here as shown in Figure R3 (*ACS Nano*, 2018, 12, 10262-10271). Figure 4b is the voltage curves of the main TENG with a $10 M\Omega$ external load under 5 Hz working frequency as shown in Figure R2b.

Figure R3. The testing circuit of voltage in Fig. 3a.

8. Please double check the conclusion component. Several “space” need to be deleted in this component. For example, line 263 and line 267.

Answer: Thank the reviewer for this detailed comment. We have revised the conclusion component accordingly, and recheck the whole manuscript carefully.

Reviewer #2 (Remarks to the Author):

This review relates to the Manuscript Number NCOMMS-19-195864 submitted to “Nature Communications” journal entitled, “Quantification of surface contact efficiency and air breakdown model on charge excitation TENG for maximum charge density.”

The manuscript reports on an interesting strategy to achieve high charge density based on Paschen’s law, which is intended to improve the output performance of TENGs. While premise is certainly interesting, there are major questions that are not fully addressed, and publications at this stage seems a bit premature.

Answer: We appreciate the reviewer’s detailed and responsible comments on our work. In this work, we built the maximum charge density model for our recently developed charge excitation TENG (CE-TENG) based on Paschen’s Law and capacitive mode, and further analyzed and discussed the strategy of improving charge output, and finally we achieved the recorded high output charge density experimentally. This work would give some effective instructions to the further charge output enhancement of TENG devices. We believe that the significance of this work in TENG field deserves to be published on Nature Communications. We hope that after reading the response and the revised version of the manuscript, the reviewer would make a more positive decision on this work.

1. Given that the optimized thickness of PEI, 4 μm , it is not clear how this is achieved and/or controlled. Authors highlighted the contact efficiency, but there is a lack of academic/scientific description or understandable mechanism that how this thickness is associated. It would be interesting to show thickness ($< 4 \mu\text{m}$) dependence with comments about this.

Answer: We highly appreciate the reviewer for raising up this question.

Firstly, we would like to declare that all the dielectric films used in the experiment are commercialized and 4 μm PEI film is the thinnest one we can purchase from the market (response to “it is not clear how this is achieved and/or controlled”. And we have added the declaration of materials usage in the manuscript).

Secondly, in this work, we established the maximum charge density of CE-TENG by Paschen’s Law and capacitive model in theory:

$$\sigma_{max} = \left(\frac{AP\varepsilon_0}{(\ln(Px)+B) \left(1 - \frac{x}{\frac{d}{\varepsilon_r} + x + \frac{\varepsilon_0 S}{C}} \right)} \right)_{min}$$

This equation is also plotted in Fig 1f, which indicates a thinner dielectric layer would lead to a higher output charge density. In the experiment, 25 μm , 12 μm , 5 μm and 4 μm dielectric film were used and experimental results highly consistent with the theoretical expectation. The dielectric thinner than 4 μm is our further work, and the scientific problem for the thinner film is not the same as discussed in this work. (response to “It would be interesting to show thickness ($< 4 \mu\text{m}$) dependence with comments about this”. And this is the reason we use the thinnest commercialized 4 μm PEI film mostly in this work).

Thirdly, the surface roughness would form air voids, which cause insufficient contact between top electrode and bottom dielectric, and the insufficiency would become more significant when reducing the dielectric thickness which would largely affect

output charge performance. In this case, we define the contact efficiency as:

$$\eta = \frac{C_{contact}}{C_{film}}$$

Where $C_{contact}$ is the capacitance when device getting compressed, and C_{film} stands for the capacitance of dielectric film with deposited electrodes on both sides. Utilizing the definition, we theoretically analyze the contact efficiency and dielectric thickness as presented in **Supplementary Note 2**, and systematically measure the experimental values using 25 μm , 12 μm , 5 μm and 4 μm dielectric layer as shown in **Supplementary Table 1**. Moreover, we also characterize this point using 4 μm dielectric film when varying surface roughness as depicted in **Supplementary Figure 4**. In the manuscript, this point is also presented in Fig a-c and the main text. Therefore, we believe this point is well presented and discussed. (response to “Authors highlighted the contact efficiency, but there is a lack of academic/scientific description or understandable mechanism that how this thickness is associated”)

Supplementary Note 2. The effect of contact status on output with reducing the dielectric thickness

Here, we define the efficiency of contact status as the equation presented:

$$\eta = \frac{C_{contact}}{C_{film}} \quad (1)$$

Where $C_{contact}$ is the capacitance when device getting compressed, and C_{film} stands for the capacitance of dielectric film with deposited electrodes (**Supplementary Figure 7**).

Considering the existence of air void under compressed state. The following equations can be obtained:

$$C_{contact} = \frac{\epsilon_r \epsilon_0 S}{\epsilon_r h + d} \quad (2)$$

$$C_{film} = \frac{\epsilon_r \epsilon_0 S}{d} \quad (3)$$

Where S is the capacitor electrode area. ϵ_r and ϵ_0 is relative permittivity of dielectric and vacuum respectively. h represents the equivalent air gap created by air voids. d is the dielectric thickness.

Therefore, the efficiency of contact status can be expressed by:

$$\eta = \frac{d}{d + \epsilon_r h} \quad (4)$$

According to equation (4), with a constant contact status (constant h), while reducing

the dielectric thickness, the efficiency of contact would become worse. **Supplementary Table 1** shows the experimental tested results.

Supplementary Table 1. The systematical comparison of the contact status of different thickness dielectric films used in the experiment

Film thickness	Deposited capacitance (C_0)	Contact capacitance (C_1)	Contact efficiency ($\eta=C_1/C_0$)	Traditional charge density (mC/m^2)	Actual charge density (mC/m^2)
4 μm	9.96nF	5.48nF	54.98%	2.20	4.001
5 μm	6.44nF	3.21nF	49.84%	1.55	3.112
12 μm	3.10nF	2.01nF	64.84%	0.95	1.466
25 μm	1.31nF	1.21nF	92.37%	0.60	0.649

Supplementary Figure 4. The output performance of CE-TENG with different contact status formed by sandpaper-electrodes. **(a)** The photograph of the deposited

electrode based on sandpaper (electrode area: $3.2 \times 3.2 \text{ cm}^2$). **(b)** The output performance of the deposited electrodes based on sandpaper with different roughness. **(c)** SEM images of surface of deposited electrodes based on sandpaper with different roughness (#10000, #7000, #5000, #3000, #2000).

2. Authors particularly stated that “In the experiments, through our contact optimization, the contact efficiency of 4 μm dielectric can be improved...”, but it is not at all clear why authors characterized the whole system taking advantage of the external capacitance 25 nF, and the explanation provided is far from adequate.

Answer: Thanks for this detailed reviewing. In the work, according to the charge excitation TENG model equation,

$$\sigma_{max} = \left(\frac{AP\varepsilon_0}{(\ln(Px)+B) \left(1 - \frac{x}{\frac{d}{\varepsilon_r} + x + \frac{\varepsilon_0 S}{C}} \right)} \right)_{min} \quad (1)$$

Where C represents the external capacitor, we can clearly see that the maximum charge density of CE-TENG increases along with the increasing of external capacitor in theory. Experimental demonstration is also carried out, in Supplementary Figure 6, where we tested the maximum charge density under 5 nF, 10 nF, 25 nF, 50 nF and 110 nF external capacitor. It is clear that the charge density increases with the increasing of external capacitor, which is consistent with the theory. However, large external capacitor spends longer time to reach charge saturation as shown in Fig. 3b and Supplementary Figure 6b (600 s for 110 nF and 100 s for 25 nF), which is not suitable to practical applications. Therefore, on balance, we decided to adopt external capacitor with 25 nF. Related descriptions have been added into the manuscript as follows. (response to “but it is not at all clear why authors characterized the whole system taking advantage of the external capacitance 25 nF, and the explanation provided is far from adequate”)

Supplementary Figure 6a demonstrates the impact of different external capacitance on the output charge, indicating a larger external capacitor leads to a higher charge density. However, a large external capacitor spends a longer time to reach charge saturation as shown in Fig. 3b and **Supplementary Figure 6b**. Therefore, on balance, we adopt 25nF external capacitance to test the entire output performance characteristics.

Fig.3b | Dynamic charge density output of CE-TENG without using zener diode (25 nF external capacitor).

Supplementary Figure 6. The output charge density with different external capacity. (a) The output charge density under different external capacitors without voltage stability, the working frequency of the CE-TENG is 3Hz. (b) Dynamic charge density output of CE-TENG without using the zener diode (the maximum charge density can achieve 2.55 mC m^{-2} when the external capacitor is 110 nF, the working frequency and the relative humidity are respectively 3Hz and 5%).

3. It is not particularly useful to compare (in Fig. 2e) charge density as a function of bias voltage (DC) and derived charge output values generated by charge excitation TENG (AC or pulsated AC), which is not fully accurate.

Answer: Thanks for this detailed question. For charge excitation TENG, the working mechanism is equal to the combination of voltage source, external capacitor and main

TENG (*Nat. Common.* 2019,10,1426), (*Nano Energy*, 2018, 49, 625–633). And the relationship of charge density σ and excitation voltage V_{CE} is described by:

$$\sigma = C_M V_{CE} \quad (2)$$

where C_M is the contact capacitance of main TENG. Figure 2e shows the output charge density when applying different bias voltage (*Nat. Common.* 2018, 9, 3773; *Nat. Common.* 2019,10,1426). From this Figure, we can clearly see that with the voltage raising and reaching over 400 V, the output charge density linearly increases firstly and then gets the saturated due to the limitation of air breakdown effect. The purpose of this figure is to prove that the voltage provided by external TENG can make the output charge density of main TENG achieve saturated. I'm sorry that we didn't explain it clearly, we have added detailed descriptions in the manuscript as follow.

The output charge density of CE-TENG with 4 μm dielectric under a constant environment and application of different excitation voltages are shown in Fig. 2e. With the voltage raising and reaching over 400 V, the output charge density linearly increases and gets the saturated due to the limitation of air breakdown effect discussed above.

According to Fig. 2e, the open-circuit voltage of external TENG over 1000 V is enough to ensure the main TENG working near the maximum level.

Fig. 2d, Electric scheme of CE-TENG when exciting by a voltage source. 2e, Output charge density when applying different bias voltage (operation frequency: 1 Hz, relative humidity: 5 %, dielectric thickness: 4 μm).

Figure R4. Basic working mode of external-charge-excitation TENG. (Nat. Commun. 2019,10,1426)

4. What accounts for the varied charge density as a function of relative humidity in Fig. 2g? Has there been any charge density drop in optimized PEI? This seems like far more important parameter that has not been discussed at all. One of the weakest parts of the manuscript is in the characterization of the device at 5% relative humidity (RH). I would suggest a few more experimental test to be performed at different RH (moderate level).

Answer: Thanks for your good question and valuable suggestion. According to the air-breakdown equation of charge excitation TENG,

$$\sigma_{max} = \left(\frac{AP\epsilon_0}{(\ln(Px)+B) \left(1 - \frac{x}{\frac{d}{\epsilon_r} + x + \frac{\epsilon_0 S}{C}} \right)} \right)_{min}$$

A and B are the constants determined by atmosphere, including atmospheric composition, relative humidity, temperature and etc. A is inversely proportional to the relative humidity and is proportional to maximum charge density, therefore, we can enhance maximum charge density by creating low relative humidity environment to increase constant A as shown in Figure 1f and Figure 2g from theoretical and experimental aspects. (response to “What accounts for the varied charge density as a function of relative humidity in Fig. 2g?”)

From these results, a lower relative humidity leads to a larger output charge density. In our experimental condition, relative humidity of 5% is the lowest one (optimized one) we can create, so the characterizations of CE-TENG devices at 5% relative humidity were particularly used. (response to “One of the weakest parts of the manuscript is in the characterization of the device at 5% relative humidity (RH).”)

And we use the maximum charge density-humidity curve with 4 μm PEI film to replace the old Figure 2g as follow, which is consistent with theoretical result in Fig.1f.

Fig. 1f | The maximum charge density limited by air break down with different dielectric thickness and atmospheric constant A.

Fig. 2g | Output charge density of CE-TENG when varying atmospheric humidity (dielectric thickness: 4μm PEI, bias voltage: 400 V)

5. Authors studied the performance of the CE-TENG at 3 Hz of applied mechanical load, yet demonstration presenting its high output performance was measured at 5 Hz. How working frequency would be expected to function at their capacitance of the whole device?

Answer: Thank the reviewer for this valuable question. TENG is a kind of charge-determined generator, whose output performance is strictly limited by charge density (*Nat. Common.* 2015, 6, 8376). Figure R5 shows the circuit diagram to drive electric device, and the average current of CE-TENG is calculated as $I_{TENG} = 2fS\sigma$, where S and σ is the area and charge density of main TENG respectively. Only when the average current of CE-TENG is larger than or equal to the working current I_{device} of electric device, that the TENG can drive electric device successfully, hence, working frequency of CE-TENG is as follow:

$$f \geq \frac{I_{device}}{2S\sigma} \quad (2)$$

The working frequency f of TENG is depended on the output charge Q of TENG and average working current I_{device} of electric device. The working current of electric device is about 23.1 μA as shown in Figure R6, while the charge density and area of main TENG is about 2.38 mC/m² and 10 cm² respectively, thus, working frequency must be larger than 4.92 Hz if CE-TENG could drive electric device successfully. Therefore, we drive the temperature hygrometer at 5 Hz working frequency.

Figure R5. The diagram of driving temperature hygrometer with CE-TENG

Figure R6. The driving current of temperature hygrometer at a constant voltage source.

On a more general note, the manuscript seems fall short of the standards expected from a journal like Nature Communications. Unfortunately, based on the above comment, I am unable to recommend publication.

Answer: We hope that after reading the response and the revised version of the manuscript, the reviewer could make a more positive decision on this work.

Reviewer #3 (Remarks to the Author):

The charge density is one of the key indicators to evaluate triboelectric nanogenerator's performance. It has been improved from $460 \mu\text{C}/\text{m}^2$ (Nat. Communication, 2018, DOI: 10.1038/s41467-018-06045-z) to $1000 \mu\text{C}/\text{m}^2$ (Nano Energy, 49, 625-633 (2018)) to $1250 \mu\text{C}/\text{m}^2$ (Nat. Communication, 10, 1426 (2019)). This paper presents an interesting experimental and theoretical investigation on further improving charge density in a charge excitation triboelectric nanogenerator (TENG). By using a $4 \mu\text{m}$ thick PEI film and a homemade carbon/silicon gel

electrode, the authors managed to push the charge density to 2.38 mC/m², setting a new record for TENGs. Theoretical modeling predicts an even higher value. This research is of substantial interest to the scientific communities in energy harvesting devices, material science and device physics. All figures are well prepared and the experimental data are carefully analyzed. I recommend to publish this paper.

Answer: We highly appreciate the reviewer's positive and valuable comments on our work as "a new record". And we also thank the reviewer's detailed and responsible reviewing of our work.

A few more comments/questions:

1. How to set the optimum Zener diode in advance, as the offset in Zener value might deviate the optimum power?

Answer: Thanks for this valuable question. In practical, we met the same problem in this experiment, and the most difficult point is to find zener diode or zener diode group with certain parameters even though we know the suitable parameter. To get the parameter of zener diode, firstly, we should test the contact capacitor C_M of main TENG with a multimeter. It is worth note that the contact capacitor of main TENG calculated by $C = \frac{\epsilon_0 \epsilon_r}{d}$ is larger than the real value due to incomplete contact between electrodes and dielectric film, so the measured capacitance is the correct one. Secondly, we need to get the charge density-time curve (Fig 3b) of charge excitation TENG without zener diode to get the maximum charge density σ_M , and then we can get the value of excitation voltage $V_{CE} = (S \cdot \sigma_M) / C_M$, which is also the working voltage of zener diode. In practical experience, the working voltage of zener diode is setting as 1.2~1.5 V_{CE} to ensure get the maximum output charge density under voltage stabilization. Besides, the working current of zener diode under 1.2-1.5 V_{CE} is another important parameter to choose zener diode.

Finally, to achieve voltage stabilization for CE-TENG, which means that the superfluous accumulation charge must be released through the zener diode, in other word, the average accumulation charge from the external TENG is the working current of zener diode. Therefore, the working current (this work) of zener diode is $I_{CE} = Qf$ (half-wave rectifier circuit), where f is the working frequency of CE-TENG and Q is the output charge of external TENG (Response to "How to set the optimum Zener diode in advance").

In Fig. 3b and Supplementary Figure 8, we can see that the maximum effective charge density keeps stable even though the excitation voltage increases after the effective charge density reaches saturation. Thus, the offset of working voltage of zener diode toward positive direction will not deviate the optimum power (Response to “as the offset in Zener value might deviate the optimum power?”).

2. Can the authors comment on the device performance (charge accumulation, device stability) at frequencies higher than 5 Hz?

Answer: Thanks for this good question. I am sorry that we only display the output performance of CE-TENG with frequency less than or equal to 5Hz limited by the highest operating frequency of liner motor. The accumulation charge Q_A in main TENG can be described as $Q_A = tQf$, where t is the operating time, Q is the charge of external TENG and f is the operating frequency of CE-TENG, therefore, the charge accumulation speed will increase with the increase in frequency and the main TENG will reach saturation within less time. In addition, the dielectric film may damage under excessively higher operating frequencies, therefore, the device stability may decline.

3. Has the energy conversion efficiency of the reported CE-TENG been measured? If yes, what is it? Is it following the same trend as that of the charge density?

Answer: Thank the reviewer for raising up this question. Previous work reported that freestanding TENG has a 100% theoretical conversion efficiency (*Adv. Mater.* 2014, 26, 2818–2824) and the actual conversion efficiency is about 85% (*Adv. Mater.* 2014, 26, 6599–6607). We calculated the conversion efficiency of CE-TENG in this work to be about 40~58% according to the calculation method above, and the energy consumption here should be considered as the sum of all parts. The main energy consume is from external TENG and mechanical motion, therefore, the energy conversion efficiency will increase appropriately along with the increase of charge density.

4. In supporting Figure 6, the authors report a maximum charge density of 2.55 mC/m² that was achieved when using an external storage capacitor of 110 nF. This value is better than the one reported in the abstract. Shall this 2.55 mC/m² be mentioned in the abstract or in the conclusion?

Answer: Thank reviewer for the detailed reviewing. As for the air breakdown model

of charge excitation TENG, both the external capacitor and thickness of dielectric film have an influence on maximum charge density, larger external capacitor leads to higher charge density as shown in Supplementary Figure 6a.

$$\sigma_{max} = \left(\frac{AP\varepsilon_0}{(\ln(Px)+B) \left(1 - \frac{x}{\frac{d}{\varepsilon_r} + x + \frac{\varepsilon_0 S}{C}} \right)} \right)_{min} \quad (1)$$

However, large external capacitor spends longer time to reach charge saturation as shown in Supplementary Figure 6b (600 s for 110 nF and 100 s for 25 nF), which is not suitable to practical applications. Therefore, we do not mention 2.55 mC/m² charge density in abstract and introduction for displaying a more responsible and practical result.

Supplementary Figure 6. The output charge density with different external capacity. (a) The output charge density under different external capacitors without voltage stability at the working frequency of 3Hz for the CE-TENG. (b) Dynamic charge density output of CE-TENG without using the zener diode (the maximum charge density can achieve 2.55 mC m⁻² when the external capacitor is 110 nF at the working frequency of 3 Hz and the relative humidity of 5%).

5. Can the authors provide more information about the electric properties of their homemade carbon/silicon gel electrode?

Answer: Thank reviewer for this detailed reviewing. I am sorry that we only display the resistance of the homemade carbon/silicon gel electrode limited by the experimental equipment, and it is about $2.5 \sim 7.6 \times 10^{-3} \Omega \cdot m$.

6. Two typos:

- a. Line 45: “concatct”
- b. Line 55: “(CE-ENG)”

Answer: Thank the reviewer for the detailed reviewing. We have revised the manuscript accordingly.

Reviewers' Comments:

Reviewer #1:

Remarks to the Author:

This manuscript reports an advanced approach to enhance and evaluate the surface charge density of triboelectric nanogenerator (TENG), which is vitally important in TENG related energy harvesting and sensing. The authors demonstrated the highest record of average charge density up to 2.38 mC m⁻² using a 4 μm PEI film and homemade carbon/silicone gel electrodes with an ambient atmosphere of 5% relative humidity. The prospective technic approach provided in this manuscript for surface charge density enhancement will greatly push the output performance of TENG to a new horizon. The outcome of this work is impressive and is also important in the TENG field. Overall, the reply to comments is very convincing and helpful. Thus, I recommend a formal acceptance for this manuscript.

Reviewer #2:

Remarks to the Author:

The authors took into consideration the reviewers' comments, they analyzed additional measurements and answered to the questions. They addressed most of reviewers' comments and took them into account by modifying the manuscript and the supplementary information. They nicely answered the questions by providing data, analysis and discussions. The manuscript has been changed and is now better and clearer. Overall I think this work can now be published in Nature Communications. This is why I recommend accepting this manuscript.

Reviewer #3:

Remarks to the Author:

The authors have addressed all concerns and comments. I recommend to publish the paper as is.

Point-to-Point Response to the Reviewer's Comments
(Comments in black, response in blue)

Reviewer #1 (Remarks to the Author):

This manuscript reports an advanced approach to enhance and evaluate the surface charge density of triboelectric nanogenerator (TENG), which is vitally important in TENG related energy harvesting and sensing. The authors demonstrated the highest record of average charge density up to 2.38 mC m⁻² using a 4 μm PEI film and homemade carbon/silicone gel electrodes with an ambient atmosphere of 5% relative humidity. The prospective technic approach provided in this manuscript for surface charge density enhancement will greatly push the output performance of TENG to a new horizon. The outcome of this work is impressive and is also important in the TENG field. Overall, the reply to comments is very convincing and helpful. Thus, I recommend a formal acceptance for this manuscript.

Response: Thanks for your strong efforts and valuable comments on our work.

Reviewer #2 (Remarks to the Author):

The authors took into consideration the reviewers' comments, they analyzed additional measurements and answered to the questions. They addressed most of reviewers' comments and took them into account by modifying the manuscript and the supplementary information. They nicely answered the questions by providing data, analysis and discussions. The manuscript has been changed and is now better and clearer. Overall I think this work can now be published in Nature Communications. This is why I recommend accepting this manuscript.

Response: Thanks for your strong efforts and valuable comments on our work.

Reviewer #3 (Remarks to the Author):

The authors have addressed all concerns and comments. I recommend to publish the paper as is.

Response: Thanks for your strong efforts and valuable comments on our work.